# A Literature-Derived Knowledge Graph Augments the Interpretation of Single Cell RNA-seq Datasets

**DOI:** 10.3390/genes12060898

**Published:** 2021-06-10

**Authors:** Deeksha Doddahonnaiah, Patrick J. Lenehan, Travis K. Hughes, David Zemmour, Enrique Garcia-Rivera, A. J. Venkatakrishnan, Ramakrishna Chilaka, Apoorv Khare, Akhil Kasaraneni, Abhinav Garg, Akash Anand, Rakesh Barve, Viswanathan Thiagarajan, Venky Soundararajan

**Affiliations:** 1nference, One Main Street, Cambridge, MA 02142, USA; deeksha@nference.net (D.D.); patrick@nference.net (P.J.L.); travis@nference.net (T.K.H.); dzemmour@nference.net (D.Z.); enrique@nference.net (E.G.-R.); aj@nference.net (A.J.V.); 2nference Labs, Bengaluru, Karnataka 560017, India; ramakrishna@nference.net (R.C.); apoorvkhare@nference.net (A.K.); akhilkasaraneni@nference.net (A.K.); abhinav@nference.net (A.G.); akash@nference.net (A.A.); rbarve@nference.net (R.B.); vishy@nference.net (V.T.)

**Keywords:** single cell genomics, natural language processing

## Abstract

Technology to generate single cell RNA-sequencing (scRNA-seq) datasets and tools to annotate them have advanced rapidly in the past several years. Such tools generally rely on existing transcriptomic datasets or curated databases of cell type defining genes, while the application of scalable natural language processing (NLP) methods to enhance analysis workflows has not been adequately explored. Here we deployed an NLP framework to objectively quantify associations between a comprehensive set of over 20,000 human protein-coding genes and over 500 cell type terms across over 26 million biomedical documents. The resultant gene-cell type associations (GCAs) are significantly stronger between a curated set of matched cell type-marker pairs than the complementary set of mismatched pairs (Mann Whitney *p* = 6.15 × 10^−76^, r = 0.24; cohen’s D = 2.6). Building on this, we developed an augmented annotation algorithm (single cell Annotation via Literature Encoding, or scALE) that leverages GCAs to categorize cell clusters identified in scRNA-seq datasets, and we tested its ability to predict the cellular identity of 133 clusters from nine datasets of human breast, colon, heart, joint, ovary, prostate, skin, and small intestine tissues. With the optimized settings, the true cellular identity matched the top prediction in 59% of tested clusters and was present among the top five predictions for 91% of clusters. scALE slightly outperformed an existing method for reference data driven automated cluster annotation, and we demonstrate that integration of scALE can meaningfully improve the annotations derived from such methods. Further, contextualization of differential expression analyses with these GCAs highlights poorly characterized markers of well-studied cell types, such as CLIC6 and DNASE1L3 in retinal pigment epithelial cells and endothelial cells, respectively. Taken together, this study illustrates for the first time how the systematic application of a literature-derived knowledge graph can expedite and enhance the annotation and interpretation of scRNA-seq data.

## 1. Introduction

The development of single cell transcriptomic technologies has enabled the dissection of cellular heterogeneity within complex tissue environments [1,2,3,4,5]. Typically, the processing workflows for such studies involve unsupervised clustering of single cells based on their gene expression profiles followed by the assignment of a cell type annotation to each identified cluster. While cell type annotation was initially performed via manual inspection of cluster-defining genes (CDGs), there have been a number of algorithms developed recently to automate this process [6,7,8,9,10,11,12,13].

Manual cell type annotation inherently relies on an individual’s knowledge of cellular gene expression profiles. For example, an immunologist may know based on their literature expertise and firsthand experience that CD19 and CD3E are specific markers of B cells and T cells, respectively. On the other hand, the existing methods for automated annotation leverage previously generated transcriptomic datasets or curated lists of cell type-defining genes to determine which cell type a newly identified cluster most closely resembles. Interestingly, it is now common for researchers to employ both manual and automated methods for cluster annotation [14,15,16,17], as one can be used to check the veracity of the other. Importantly, there remains an unmet need for tools which augment manual annotation by transparently and objectively leveraging the associations between genes and cell types which are embedded in the literature.

Rare or novel cell types which are marked by well-studied genes have been identified through scRNA-seq, such as the CFTR-expressing pulmonary ionocyte described recently [18,19]. Conversely, scRNA-seq should also enable researchers to identify novel markers of even well characterized cell types. However, there is currently no standardized method to assess the level of literature evidence for each individual CDG during the process of manual or automated cell type assignment. As a result, this step is often foregone in practice, in favor of proceeding to downstream workflows such as differential expression, pseudotime projection, or analysis of receptor-ligand interactions.

Here, we leverage a literature-derived knowledge graph to augment the annotation and interpretation of scRNA-seq datasets. We first deploy an NLP framework to quantify pairwise associations between human protein-coding genes and cell types throughout the biomedical literature contained in PubMed. We validate that these quantified gene-cell type associations (GCAs) recapitulate canonical cell type defining genes and then harness them to perform unbiased literature-driven annotation of scRNA-seq datasets. Finally, we demonstrate that integration of these literature encoded GCAs into a differential expression workflow can highlight unappreciated gene expression patterns that warrant further experimental evaluation.

## 2. Methods

### 2.1. Generation of Cell Type Vocabulary

To create a database of cell types, we first manually curated over 300 cell types into a directed acyclic graph structure. Specifically, each unique cell type corresponds to one node in the graph, which can be connected to one or more parent nodes (i.e., broader cell type categories) and one or more child nodes (i.e., more granular subsets of the given cell type). For example, “T cell” corresponds to one node, of which “lymphocyte” is a parent node and “CD4^+^ T cell” is a child node. Where applicable, we also manually added aliases or acronyms for each cell type. We merged this manually curated cell graph with the EBI Cell Ontology graph [20,21,22] by mapping identical nodes to each other and preserving all parent child relationships documented in each graph. This merging was needed because there are various cell types which have been identified in the scRNA-seq datasets that we used in this study (e.g., cell types defined by markers or functional states) which have not yet been incorporated into the Cell Ontology graph. Further, there are some instances in which relevant edges are missing from Cell Ontology (e.g., there is no relationship captured between “foam cells” and “macrophages”).

We expanded each node to include synonymous tokens by considering synonyms from the EBI Cell Ontology and Unified Medical Language System database [23] and from our own custom alias identification service. After expansion, we reviewed our updated graph to remove erroneously introduced synonyms and performed an internal graph check to ensure that each token (i.e., cell type name or synonym thereof) occurred in one and only one node. The complete cell graph is given in Appendix A.

For the cell type annotation algorithm described subsequently, this graph was filtered to retain the 556 nodes which were strongly associated (local score ≥ 3; see description of local scores below) with at least one human protein-coding gene; these 556 nodes are subsequently referred to as “candidate cell types”. We also defined a set of 104 “priority nodes” which intend to capture major cell types or cell type categories, to which all other candidate cell types in the filtered graph are mapped. The set of all candidate cell types, along with the priority nodes to which they map, are given in Appendix A (indexed by candidate cell type) and Appendix A (indexed by priority node).

### 2.2. Generation of Gene Vocabulary

We obtained the full set of human protein-coding genes from HGNC [24] and curated potential gene synonyms from various sources including ENTREZ, UniProt, Ensembl, and Wikipedia. For specific gene families, we also manually added family-level synonyms which are not captured by synonyms curated at the single gene level. This included genes encoding the following proteins: T cell receptor subunits, immunoglobulin subunits, class II MHC molecules, hemoglobin subunits, surfactant proteins, chymotrypsinogen subunits, CD8 subunits (CD8A, CD8B), and CD3 subunits (CD3E, CD3G, CD3D, and CD247). The complete gene vocabulary is given in Appendix A.

For the cell type annotation algorithm described subsequently, we only considered protein-coding genes which were strongly associated with at least one cell type in the literature (local score ≥ 3; see description of local scores below). Further, we excluded mitochondrially encoded genes (gene names starting with “MT-”), genes encoding ribosomal proteins (gene names starting with “RPS”, “RPL”, “MRPS”, or “MRPL”), and MHC class I genes except for HLA-G (HLA-A, HLA-B, HLA-C, HLA-E, HLA-F). These filtering steps yielded a final set of 5113 “eligible genes” for consideration during the cell type annotation steps (indicated in Appendix A).

### 2.3. Quantification of Literature Associations between Genes and Cell Types

To quantify gene-cell type associations (GCAs) in biomedical literature, we computed local scores as described in detail previously [25]. Briefly, this metric measures how frequently two tokens *A* and *B* are found in close proximity to each other (within 50 words or fewer) in the full set of considered documents (corpus), normalized by the individual occurrences of each token in that corpus. In this case, the two tokens are a gene and a cell type, and the corpus includes all abstracts in PubMed along with all full PubMed Central (PMC) articles.

To calculate the score, we first compute the pointwise mutual information between *A* and *B* as pmi*_AB_* = log_10_([Adjacency_AB_ * N_C_]/[N_A_ * N_B_]), where Adjacency_AB_ is the number of times that Token A occurs within 50 words of Token B (or vice versa), N_A_ and N_B_ are the number of times that Tokens A and B each occur individually in the corpus, and N_C_ is the total number of occurrences of all tokens in the corpus. The local score between Tokens A and B is then calculated as LS_AB_ = ln(Adjacency_AB_ + 1)/[1 + e^−(pmiAB−1.5)^]. A local score of 0 indicates that Tokens A and B have never occurred within 50 words of each other, and a local score of 3 indicates a co-occurrence likelihood of approximately 1 in 20. Thus, we typically consider a local score greater than or equal to 3 to represent a significant literature association.

A matrix of GCAs (i.e., pairwise local scores between all genes and all candidate cell types) is provided in Appendix A. Notably, the maximum GCA varies substantially across the set of candidate cell types (range 3.00–12.54; see Appendix A), which reflects the fact that different cell types have been characterized to different degrees with respect to the landscape of all human genes. To account for this, we also computed “scaled GCAs” by dividing all GCAs for a given cell type by the maximum GCA for that cell type, such that the maximum scaled GCA for every cell type is equal to 1. The matrix of scaled GCAs is provided in Appendix A.

### 2.4. Curation of Canonical Cell Type Defining Genes

To test the utility of literature-derived GCAs in capturing gene expression profiles, we first manually curated a set of cell type defining genes that were indicated by authors to have been used during cell type annotation in previously published manually annotated scRNA-seq datasets [26,27,28,29,30,31,32,33]. The complete set of manually curated cell type defining genes is provided in Appendix A. We also obtained a previously curated set of cell type defining genes from the Panglao database [34]. From this database, genes were extracted that were labeled as canonical human markers with a ubiquitousness index < 0.06, human sensitivity > 0, and mouse sensitivity > 0 (Appendix A).

Using these curated sets of defining genes, all pairwise gene-cell type combinations and their corresponding local scores (GCAs) were then classified as “matched” or “mismatched”. A matched GCA refers to the local score between a cell type and one of its defining genes (e.g., B cells and CD19); a mismatched GCA refers to the local score between a cell type and any non-defining gene for that cell type (e.g., B cells and CD3E). That is, the mismatched pairs are obtained by simply excluding matched pairs from the set of all unique pairwise combinations of the genes and cells contributing to the matched pairs. From the manually curated set, there were 174 matched pairs and 5678 mismatched pairs; from the Panglao database, there were 2291 matched pairs and 154,313 mismatched pairs.

For the manually curated set, we found that each cell type had at least one “matched” gene with a local score ≥ 3. To depict this finding and to illustrate that generally a gene that is strongly associated with one cell type is not strongly associated with most others, we plotted a heatmap of local scores in which the dimensions were cell types and genes, where one gene with a local score ≥ 3 was selected for each cell type.

### 2.5. ROC Analysis of Local Scores to Classify Matched GCAs versus Mismatched GCAs

To evaluate the ability of our literature-derived GCAs (local scores) to classify gene-cell type pairs as matched or mismatched, we performed a Receiver Operating Characteristic (ROC) analysis and computed the area under the ROC curve (AUC) using the “pROC” package (version 1.17.0.1) in R (version 4.0.3). Briefly, we tested over 500 thresholds (sliding intervals of 0.01 starting at 0) to determine the sensitivity and specificity of local scores in classifying gene-cell type pairs as matched or mismatched. We also repeated this ROC analysis 10,000 times with randomly shuffled assignments of “matched” and “mismatched” to empirically confirm a lack of predictive power for local scores in performing this classification with the given sets of genes and cell types when labeled at random.

### 2.6. Processing of scRNA-seq Studies

For each individual human scRNA-seq study listed in Appendix A [26,27,28,29,30,33,35,36,37,38,39,40,41,42,43,44,45,46,47,48,49,50,51,52,53,54,55,56,57,58,59,60,61,62,63,64,65,66,67,68], we obtained a counts matrix and metadata file (if available) from the Gene Expression Omnibus or another public data repository. We then processed each dataset using Seurat v3.0 [69]. Cells with fewer than 500 counts or with more than 10% of counts mapping to mitochondrial genes were excluded. Doublets were excluded using scrublet (double score > 0.3) [70]. Normalization was performed using the NormalizeData function, with the method set to “LogNormalize” and the scale factor set to 10,000, such that unique molecular index (UMI) counts were converted into values of counts per 10,000 (CP10K). Scaling was performed for all genes using the ScaleData function. The top 2000 most variable genes were identified using the FindVariableFeatures function, and these genes were used as input to the subsequent steps. Linear dimensionality reduction was performed using principal component analysis (PCA), and then clusters were identified using the FindNeighbors and FindClusters functions. The top principal components explaining at least 90% of the variance were used in the FindNeighbors function, and various cluster resolutions (0.25, 0.5, 1.0, 1.5, 2.0) were tested in the FindClusters function. Cell type annotations were obtained from associated metadata files if available; otherwise, annotation was performed manually, guided by the cell types reported in the associated publication.

### 2.7. Cell Type Annotation Algorithm

To perform automated annotation of a given cluster identified from a scRNA-seq dataset using our literature-derived knowledge graph, we established an algorithm (single cell Annotation via Literature Encoding, or scALE) following these steps: (1) identify the top N cluster defining genes (CDGs); (2) compute local scores between these CDGs and all candidate cell types; (3) compute local score vector norms for each candidate cell type; and (4) rank candidate cell types for annotation plausibility based on their vector norms. Each of these steps is described in further detail in the Appendix A.

At each step, we tuned one or more hyperparameter settings including: (1) the reference data used to compute fold change values, which were used to rank candidate CDGs (within study, within tissue, or pan-study); (2) the number of CDGs used to compute GCA vector norms (1, 3, 5, 10, or 20); (3) the weighting metric used in calculating GCA L2 vector norms (fold change, log_2_FC, or unweighted); (4) the GCA version used to calculate L2 vector norms (raw or scaled); and (5) the metric used to rank the outputted cell type predictions (L2 norm, modified L0 norm, or composite rank). Each tuning setting is described in further detail in the Appendix A. In total, we tested 195 combinations of hyperparameter settings.

### 2.8. Optimization and Evaluation of Cell Type Annotation Algorithm

For each cluster C_A_ and a given combination of hyperparameter settings (“mode”), scALE outputs a table which ranks all 556 candidate cell types, from most likely annotation to least likely annotation. We used our cell graph to map the true identity of C_A_ to its priority node (“true priority node”), and each candidate cell type was similarly mapped to its priority node (“candidate priority node”). A cell type annotation was considered correct if the true priority node was the same as the candidate priority node, and we determined the success (or failure) of labeling the given cluster by identifying the minimum (best) rank at which this was the case.

We first applied scALE to a set of 13 dedicated tuning studies (containing a total of 185 clusters from 7 human tissues) to optimize the algorithm settings. The tuning studies are provided in Table 1 [26,27,28,29,30,33,35,36,37,38,39,40,41]. For each mode, we first determined the fraction of clusters (out of 185) for which the correct cell type annotation was present in the top K predictions, where K ranges from 1 to 104 (the total number of cell type priority nodes, as defined previously). For real-world application by investigators, we reasoned that an annotation would be most useful if the correct cell type was given with the top prediction, and that an annotation could also be considered useful if the correct cell type was present among the top five predicted cell types.

To account for this definition of performance, we generated a subset of the cumulative distribution plot for each parameter combination, illustrating the fraction of clusters annotated correctly within a given rank (ranging from rank 1 to rank 5). We then estimated the area under this curve (denoted as AUC_Ranks 1–5_) by calculating the average fraction of correct cluster annotations within the top 1, 2, 3, 4, and 5 ranks. Parameter combinations which yielded the highest fraction of correctly annotated clusters and the highest AUC_Ranks 1–5_ were taken as the optimal algorithm settings.

To validate the algorithm, we then applied scALE to a set of nine dedicated testing studies (containing a total of 133 clusters from eight human tissues), using the optimized combination of hyperparameter settings from our tuning experiments: (1) use pan-study reference, (2) consider top 20 CDGs, (3) weight GCAs by log_2_FC when computing L2 vector norms, (4) use scaled GCAs when computing L2 vector norms, and (5) rank final outputs by L2 norm. As above, we determined the fraction of correctly annotated clusters and the AUC_Ranks 1–5_, and we compared these values to those obtained when scALE was applied to the tuning studies. The testing studies are provided in Table 2 [71,72,73,74,75,76,77,78,79].

### 2.9. Application of scALE to Annotate to Murine scRNA-seq Data

To determine whether scALE can be applied to murine scRNA-seq data, we attempted to annotate 143 clusters identified across 18 tissues in the Tabula Muris dataset [80]. As above, we used the optimized combination of hyperparameter settings without any species-specific adjustments to the algorithm. Murine datasets contributing to the pan-study reference are provided in Appendix A [19,81,82,83,84,85].

### 2.10. Comparison of Cell Type Annotations from scALE versus SingleR

To benchmark the performance of scALE against existing automated annotation methods, we used SingleR [6] to annotate 142 clusters in ten human datasets which were labeled by scALE as part of our initial set of tuning studies [26,27,28,35,36,37,38,39,40,41]. We ran SingleR using the Blueprint Encode reference data from the celldex package in R [6,86], with the method set to “cluster”, fine.tune set to “TRUE”, and sd.thresh set to 1. The “pruned.labels” output was taken as the final annotation; if no pruned label was supplied, then the cluster was deemed to have not been annotated correctly as this indicates that the initial label was not considered robust after further pruning.

To compare the outputs from scALE and SingleR, we manually reviewed the labels to determine whether each cluster was annotated accurately by the top-ranked prediction from each method. Because the reference dataset used for SingleR inherently limits the granularity of certain annotation categories (e.g., not all epithelial or neuronal cell types are represented), we did consider these higher-level classifications as correct when appropriate. That said, we also manually reviewed each pair of labels to identify cases in which the scALE output provided helpful context to a less granular output from SingleR. For example, if SingleR labeled a cluster of hepatocytes as “epithelial cells” while scALE labeled the same cluster as “hepatocytes”, we concluded that SingleR had correctly annotated the cluster but that scALE provided additional important context to the annotation. We ultimately compared the performance by calculating the percentage of clusters which were annotated correctly by each method. We assessed the potential utility of using both methods together by calculating the percentage of correct SingleR annotations which could have been improved (e.g., made more granular) by scALE, and by identifying clusters which were annotated correctly by scALE but not SingleR, or vice versa.

### 2.11. Identification of Poorly Characterized Cell Type Markers

To identify potential novel or poorly characterized markers of well-studied cell types, we compared the mean expression (CP10K) of all genes in the defined cell type of interest to their mean expression in all other cells from our reference dataset (see Appendix A). The cell types considered here included retinal pigment epithelial cells derived from two independent human studies [30,68] and endothelial cells derived from 31 human studies [26,27,28,30,33,35,36,37,38,39,40,42,43,46,47,48,49,50,51,52,53,55,56,62,64,67,72,73,74,75,76,78,79]. Specifically, we calculated the FC value for each gene as described above in the “pan-study” method for CDG identification during the cell type annotation algorithm. We also computed the cohen’s D (*d*) as a measure of effect size which considers the variation in the two compared groups. Specifically, this was calculated as *d* = (Mean CP10K_A_ − Mean CP10K_B_)/SD_pooled_. The pooled standard deviation was calculated as SD_pooled_ = sqrt([(N_A_ − 1) × SD_A_^2^ + (N_B_ − 1) × SD_B_^2^]/[N_A_ + N_B_ − 2]), where SD_A_ and SD_B_ are the standard deviations of each individual group, and N_A_ and N_B_ are the number of single cells in each group. Group A corresponds to the cell type of interest, and Group B corresponds to all other cells contained in the reference dataset. Among genes with *d* > 0.5, we considered the top 50 genes (ranked by fold change) as cell type markers.

After identifying the cell type markers based on transcriptional data, we assessed the literature evidence relating each marker (gene) to the cell type of interest by extracting the corresponding GCAs (local scores) from Appendix A. Genes were classified as having Strong (local score ≥ 3.0), Intermediate (local score ≥ 1 and <3), or Weak (local score < 1) association to the cell type of interest.

### 2.12. Computation of Endothelial Gene Signatures

After identifying potential markers of endothelial cells that have not been well characterized, we evaluated their expression levels in several datasets. To verify the given annotation of one or more clusters in each dataset as endothelial cells, we computed an endothelial signature score for each individual cell, defined as the geometric mean of CP10K values for a selected set of canonical endothelial markers; that is, Endothelial Signature = (CP10K_Gene 1_ × CP10K_Gene 2_ × … CP10K_Gene N_)^1/N^. The five endothelial markers considered were CD31 (PECAM1), AQP1, VWF, PLVAP, and ESAM. For liver sinusoidal endothelial cells, which are known to display a distinct expression profile compared to other endothelial cells, we considered CLEC4G and CLEC4M as markers rather than the previously listed set.

### 2.13. Statistical Analysis

Statistical analyses were performed in R (version 4.0.3, R Core Team, Vienna, Austria) [87]. To compare local scores (GCAs) between matched and mismatched gene-cell type pairs, we computed a *p*-value and effect size using the two-sample Mann Whitney U Test. *p*-values were calculated using the “wilcox.test” function from the “stats” package (version 4.0.3), and effect sizes were calculated using the “wilcoxonR” function from the “rcompanion” package (version 2.3.27). This nonparametric test was applied because the mismatched GCAs did not show a normal distribution (Appendix A). We also computed cohen’s D as a measure of effect size between these groups using the “cohens_d” function from the “effectsize” package (version 0.4.3) in R. Finally, ROC curves were generated (and corresponding AUC values calculated) to assess the ability of local scores to classify matched and mismatched GCAs as described above using the “pROC” package (version 1.17.0.1).

## 3. Results

### 3.1. A Literature Derived Knowledge Graph Recapitulates Canonical Gene-Cell Type Associations

We have previously described the application of a knowledge graph trained on over 100 million biomedical documents to contextualize the scRNA-seq expression profile of ACE2, the entry receptor for SARS-CoV-2 [25]. Here we leverage this knowledge graph to comprehensively quantify gene-cell type associations (GCAs) across scientific publications. Specifically, we quantified each GCA as the local score between a single gene *G* and a single cell type *C*, which captures the likelihood of the observed co-occurrence frequency of *G* and *C* across these documents (see Section 2.3 and Figure 1A) [88]. Local scores for all such pairs are provided in Appendix A. Of note, this metric does not account for sentiment and so will capture co-occurrences regardless of whether they denote, explicitly or implicitly, the expression of a gene in a given cell type (Figure 1A). Despite this potential shortcoming, we hypothesized that local scores would generally be higher for matched gene-cell type pairs (pairs of genes and cell types in which the gene is a canonical marker for the given cell type) than for mismatched pairs (pairs of genes and cell types in which the gene is not a canonical marker for the given cell type).

To test this hypothesis, we curated matched and mismatched pairs by extracting the author-provided cluster defining genes (CDGs) that were used to classify cell types in seven manually annotated scRNA-seq studies [26,27,28,29,30,31,32,33]. This yielded 174 matched gene-cell type pairs (comprising 133 unique genes and 44 unique cell types) from tissues including blood, pancreas, lung, liver, placenta, and retina (Appendix A). We found that indeed all cell types surveyed had a strong local association with at least one of their canonical marker genes (Figure 1B). For example, T cells, B cells, pancreatic β cells, hepatocytes, and trophoblasts are strongly associated with genes including CD3E, CD20, insulin, albumin, and HLA-G, respectively (Figure 1B). Local scores between matched pairs were indeed significantly higher than local scores between mismatched pairs (Mann Whitney *p* = 6.15 × 10^−76^, r = 0.24; cohen’s D = 2.6; Figure 1C). Further, among the considered set of gene-cell type pairs, local scores were strongly predictive of whether a gene should be considered a canonical marker for a given cell type (AUC = 0.903; Figure 1D). We confirmed that local scores showed no predictive power when the matched and mismatched assignments were randomly shuffled (Appendix A).

To confirm that these observations extend beyond the cell types and tissues captured in the studies that we curated, we performed a similar analysis to compare local scores between matched and mismatched gene-cell type pairs from the Panglao database of cell type markers [34]. From this database, we extracted 2291 matched gene-cell type pairs (comprising 94 unique cell types and 1666 unique genes) and 154,313 mismatched gene-cell type pairs (Appendix A). Local scores for matched pairs were indeed significantly higher than those for mismatched pairs (Mann Whitney *p* < 1 × 10^−323^, r = 0.17; cohen’s D = 2.79; Appendix A), and local scores robustly distinguished matched from mismatched pairs (AUC = 0.88; Appendix A). We again confirmed that local scores showed no predictive power when the matched and mismatched assignments were randomly shuffled (Appendix A).

### 3.2. Literature Associations Facilitate Augmented Annotation of Single Cell RNA-seq Datasets

Having demonstrated its ability to capture canonical cell type defining genes, we hypothesized that our knowledge graph can be applied in scRNA-seq analyses to assist in automated and unbiased cluster annotation. We thus developed an algorithm (single-cell Annotation via Literature Encoding, or scALE) to annotate scRNA-seq datasets which have been clustered using any method of choice (see Section 2.7 and Figure 2). We first performed a systematic tuning analysis of scALE (see Section 2.8 and Figure 3), evaluating the impact of all 195 hyperparameter setting combinations on classification accuracy for 185 clusters from 13 previously annotated scRNA-seq datasets from seven human tissues: pancreas, retina, blood, lung, liver, kidney, and placenta [26,27,28,29,30,33,35,36,37,38,39,40,41].

This tuning analysis showed that scALE performed optimally with the following settings: (1) use the pan-study reference to identify CDGs, (2) consider the top 20 CDGs, (3) use scaled local scores to compute L2 norms, (4) weight local scores by log_2_FC to compute L2 norms, and (5) use the L2 norm alone when ranking cell type predictions (Figure 4A,B; Appendix A). With these parameters, we were able to accurately categorize 123 of 185 (66%) cell types, and the correct annotation was among the top three or five predictions for 156 (84%) and 173 (94%) of 185 cell types, respectively (AUC_Ranks1–5_ = 0.83; Figure 4C; Appendix A). Generally, using fewer than five cluster defining genes or ranking cell type predictions by the modified L0 norm were most detrimental to algorithm performance, while the reference cells used to compute CDGs, the scaling of GCAs, and the weighting of GCAs when calculating L2 norms had less impact (Figure 4A,B and Figure 5). The predicted annotations for all clusters from these tuning studies using our optimized algorithm parameters are shown in Appendix A and reviewed in the Appendix A.

We then applied scALE to a set of 9 dedicated testing studies from eight tissues (Table 2) [71,72,73,74,75,76,77,78,79]. Using the optimized set of parameters, we accurately categorized 78 of 133 (59%) clusters, with the correct annotation among the top three or five predictions for 110 (83%) and 121 (91%) clusters, respectively (AUC_Ranks1–5_ = 0.79) (Figure 6A). We further confirmed that an unmodified version of scALE performed with similar accuracy in the annotation of murine data, as 81 of 143 (57%) clusters from the Tabula Muris dataset were annotated correctly and the correct annotation was present among the top five predictions for 128 (90%) clusters (AUC_Ranks1–5_ = 0.77) (Figure 6B).

Various tools already exist to automate the cluster annotation process, so it is important to benchmark the performance of scALE against such methods. We compared the cluster labels predicted by scALE to those predicted by SingleR [6] for ten human studies [26,27,28,35,36,37,38,39,40,41]. scALE accurately categorized 115 of 142 clusters (81%), while SingleR did so for only 102 clusters (72%) (Figure 6C). The accuracy of scALE reported here is slightly higher than that observed during the previous tuning and testing because there was some additional leniency in classifications afforded due to a lack of granular epithelial cell data in the reference set used for SingleR. Thus, for this comparison, a label of “epithelial cells” could be considered correct, whereas during the tuning and testing, a more specific label (e.g., proximal tubule cell) would have been required. Importantly, we found that scALE provided context that could be used to improve (i.e., increase the granularity of) 16 of the 102 (16%) correct cluster annotations from SingleR (Appendix A). For example, while SingleR labeled loop of henle cells, hepatocytes, and pancreatic acinar cells simply as “epithelial cells”, scALE provided the correct granular annotation for each one. Taken together, this indicates that scALE performs at least as well as SingleR and could even be integrated with it to improve the current output.

### 3.3. The Literature Knowledge Graph Highlights Uncharacterized Markers of Established Cell Types

The GCAs derived from our knowledge graph can also be leveraged to rapidly and objectively assess the literature support for genes that appear to be cell type markers from any given dataset. For example, we assessed the literature evidence for the top 50 genes overexpressed in retinal pigment epithelial (RPE) cells relative to all other cells in our reference dataset (Figure 7A). Several of these genes were established RPE markers such as RPE65 and BEST1, mutations in both of which can cause retinitis pigmentosa and other retinopathies [89,90]. Other markers showing moderate to strong literature associations with the RPE included genes involved in vitamin A metabolism (e.g., TTR, RLBP1, RBP1, LRAT). However, we also identified several CDGs with little or no literature association to RPE cells, including genes encoding ion transporters (SLC6A13, CLIC6) and proteins that modulate Wnt signaling (FRZB, SFRP5). While these Wnt modulators have been infrequently referenced as RPE markers [91,92], and CLIC6 has been detected by proteomics in RPE tissue and cells [93,94], the contribution of these genes to RPE function has never been explored. We confirmed that these markers were also upregulated in RPE cells relative to other retinal cells using the more traditional within-study comparison as well (Appendix A).

Similarly, we assessed the existing literature evidence for genes overexpressed in endothelial cells (ECs). While 24 of the top 50 genes were strongly associated with ECs (e.g., PECAM1, VWF, ICAM1), several other genes were poorly characterized or previously uncharacterized endothelial markers (Figure 7B). For example, DNASE1L3 was identified as an EC marker whereas it is canonically reported to be expressed by macrophages and dendritic cells [95,96,97]. While its expression in liver sinusoidal ECs, non-sinusoidal hepatic ECs, and renal ECs of the ascending vasa recta has been recently reported [55,98,99,100], we not only confirmed expression in these populations (Appendix A) but also identified several other tissues in which ECs were the predominant DNASE1L3-expressing cell type, including the adrenal gland, lung, and nasal cavity (Appendix A). Further, the functional significance of DNASE1L3 expression in ECs has not been explored, but it may indeed be very relevant given the strong genetic associations connecting DNASE1L3 to the development of anti-dsDNA antibodies and various autoimmune phenotypes including lupus [96,97,101], systemic sclerosis [102], scleroderma [96,101], and hypocomplementemic urticarial vasculitis syndrome [103].

## 4. Discussion

scRNA-seq has enabled the characterization of cellular heterogeneity at unprecedented levels. Initial uptake was relatively slow due to the costs associated with these experiments, but rapid technological advances have drastically improved the accessibility of this technique [104]. As a result, the amount of scRNA-seq data which is being generated and deposited into public databases is likely to continue increasingly rapidly for years to come. It is imperative that investigators are equipped to efficiently annotate and analyze these datasets such that the insights embedded within them are not left untapped.

The cell type annotation tools developed in recent years provide excellent resources that will assist in efforts to analyze individual datasets and to synthesize the exponentially increasing quantity of publicly available data [6,7,8,9,10,11,12,13]. However, these methods do have intrinsic shortcomings, such as the requirement for the existence or generation of high quality reference transcriptomic datasets for all cell types that may be recovered in a given scRNA-seq experiment. By considering all cell types that have been described in the literature, our method circumvents this need for data curation. Further, even as these tools are increasingly applied to automate data processing pipelines, it remains preferable to perform some degree of manual review to ensure the validity of the resultant annotations. Specifically, the quality of annotations should minimally be assessed by determining whether a set of CDGs is biologically consistent, per canonical knowledge, with the given cell type label. Our method is specialized to perform this exact task.

Standard manual annotation is flawed due its inherent subjectivity and reliance on the existing knowledge of a single investigator. These flaws can be mitigated to some degree by “assisted” manual annotation that leverages search engines (e.g., Google, PubMed) to identify cell types which have been reported to express the genes under consideration. However, this process only leverages a sliver of the information contained within the biomedical literature and would be prohibitively time consuming if scaled to the analysis of large numbers of datasets. On the other hand, the deployment of NLP algorithms to mimic manual annotation provides a superior alternative, enabling “augmented annotation” workflows that objectively harness the entire knowledge graph of GCAs contained in the literature to suggest which cell types are most strongly associated with a given set of CDGs. By integrating this approach with already existing annotation workflows, one can rapidly assess the veracity of predicted cell type annotations and highlight those which warrant further review by an expert.

The utility of this literature knowledge graph also extends beyond the annotation of cell types, unlocking a new analytic workflow in the interpretation of scRNA-seq data. By its very nature, scRNA-seq provides investigators with data at the level of gene-cell type pairs, i.e., Gene *X* is expressed in Cell Type Y at Level Z. However, to date there has been no systematic and objective method to comprehensively contextualize this data with respect to the world’s knowledge of those same gene-cell type pairs at that point in time. To address this unmet need, we demonstrated that our database of literature-derived GCAs can be used to assess the degree of literature evidence connecting any gene to any cell type in which its expression has been observed by scRNA-seq. This workflow will provide investigators with the opportunity to rapidly identify gene expression patterns which were previously unknown or poorly characterized, and thereby prioritize candidates for directed follow-up functional studies.

For example, we found CLIC6 and DNASE1L3 as uncharacterized markers of RPE cells and endothelial cells, respectively. Although CLIC6 was identified almost two decades ago as a member of the chloride intracellular channel (CLIC) gene family, its function remains essentially unknown [105,106]. Biochemical studies have demonstrated that CLIC6 interacts with dopamine D2 receptors in the brain, but it is still unclear whether CLIC6 modulates dopamine receptor mediated signaling or even serves as a functional chloride channel [105,107]. It is intriguing that CLIC4, another member of the CLIC family, functions in RPE cells to promote their epithelial morphology, maintain their attachment to the photoreceptor layer of the retina, and regulate extracellular matrix degradation at focal adhesions [108,109]. CLIC6, on the other hand, has never been functionally characterized in these cells despite having been detected in them by proteomics and immunohistochemistry [93,94]. Given recent evidence for structural conservation between CLIC6 and other CLIC family proteins [110], we suggest that directed studies to analyze CLIC6 function in RPE cells are warranted.

DNASE1L3 is an extracellular DNase which is historically reported to be released specifically by macrophages and dendritic cells [95,96,101]. Our analysis challenges this canon, instead highlighting this gene as an uncharacterized marker of endothelial cells (in addition to myeloid cells) in tissues including liver, kidney, lung, nasal cavity, thyroid, and adrenal cortex. Interestingly, DNASE1L3 is genetically associated with multiple autoimmune phenotypes including systemic lupus erythematosus (SLE) [96,101], systemic sclerosis [102], and scleroderma [96,101]. Loss of function mutations in DNASE1L3 are responsible for a familial form of SLE which is characterized by the presence of anti-dsDNA antibodies and lupus nephritis [97], and mechanistic studies have directly implicated DNASE1L3 deficiency as a cause of anti-dsDNA antibody development [96]. Further, DNASE1L3 mutations have been reported to cause hypocomplementemic urticarial vasculitis syndrome, an inflammatory disease of the vascular system which often progresses to SLE [103]. Given these genetic associations, we hypothesize that the production of DNASE1L3 by ECs may protect against the development of autoantibodies and outright autoimmune disease. While CD11c^+^ cells are responsible for about 80% of serum DNASE1L3 activity in mice [96], it is plausible that ECs contribute significantly to the remaining 20%, that this fractional contribution is significantly skewed toward ECs in humans, or that EC-derived DNASE1L3 acts in a more localized fashion. Indeed, the localized activity of other EC products, such as tissue-type plasminogen activator (t-PA) and von Willebrand factor (VWF), are known to regulate clot formation specifically at sites of damaged endothelium. Perhaps the activity of DNASE1L3 in the close vicinity of renal, hepatic, and pulmonary endothelial cells protects against the development of nephritis and other forms of vasculitis at these sites.

There are several limitations of this study and the associated method (scALE). First, the GCAs which are used to perform cluster annotation simply measure literature proximity of a gene and a cell type without accounting for the sentiment surrounding this co-occurrence. scALE would be improved by training NLP models which can distinguish between co-occurrences that denote a gene expression relationship versus those that are ambiguous, spurious, or explicitly deny a gene expression relationship. Second, annotation using literature-derived associations inherently has limited utility in the recognition of novel cell types and cell types which are infrequently discussed in literature. Third, it is possible that more granular annotations can be provided by other existing pipelines (e.g., SingleR) which use reference transcriptomes of defined cellular subsets to automate the annotation process. That said, such granularity is entirely reliant on the existence and processing of high quality reference transcriptomic datasets for each subset of interest. When running SingleR with the commonly used Blueprint ENCODE reference dataset, we actually found that scALE can provide more granular annotations for epithelial and neuronal cell types. Finally, while we did demonstrate that scALE can be applied to annotate murine datasets, its performance may be improved by incorporating more species-specific considerations in future versions.

With these shortcomings in mind, we highlight that this should be viewed as a tool for augmenting current annotation workflows rather than as a standalone automated pipeline to replace other methods. Indeed, our comparison of scALE to SingleR indicated that while most clusters were correctly annotated by both methods, there were some clusters which were only classified correctly by scALE or SingleR. This suggests that formal integration of our literature-based method with data-driven annotation tools can likely increase the accuracy of annotation workflows while reducing the time and effort required for manual review.

Taken together, we have presented a new framework for the processing and interpretation of scRNA-seq datasets. Using a literature-derived knowledge graph, we comprehensively quantified the strength of associations between human genes and cell types. These associations robustly capture relationships between many cell types and their canonical gene markers, and accordingly they can be used to annotate clusters of distinct cell types identified by scRNA-seq. Finally, these associations facilitate rapid, systematic, and unbiased contextualization of lists of CDGs and differentially expressed genes, enabling investigators to identify and prioritize uncharacterized cell type markers for further exploration.

## Figures and Tables

**Figure 1 genes-12-00898-f001:**
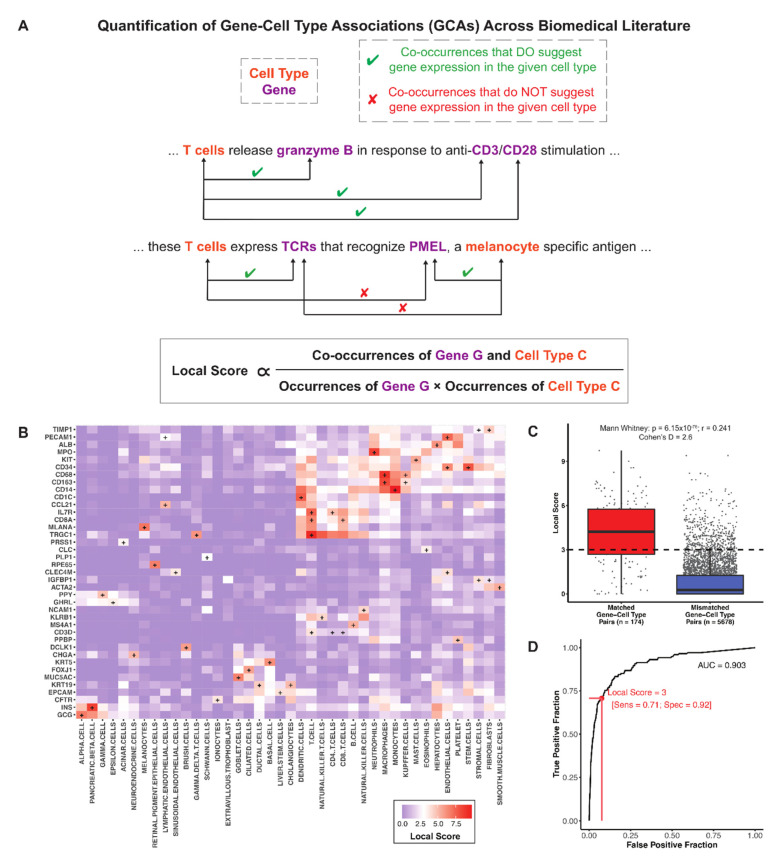
Literature-derived gene-cell type associations (GCAs) capture manually curated markers of a variety of hematopoietic, epithelial, and mesenchymal cells. (**A**) Schematic description of the computation of local scores to quantify associations between genes and cell types across biomedical literature. The local score is a proximity metric which quantifies the likelihood of the observed co-occurrence frequency of two terms within 50 words of each other. In the context of genes and cell types, some sentences with co-occurrences state or imply that a gene is expressed in a particular cell type (denoted by green check mark), while other such sentences do not (denoted by red “X”). (**B**) Heatmap depicting the pairwise local scores between cell types and corresponding cell type-defining genes. These genes and cell types were extracted from a set of scRNA-seq datasets which were previously published and manually annotated [26,27,28,29,30,31,32,33]. A “+” indicates a matched gene-cell type pair (i.e., genes which have been used to define the corresponding cell type in prior scRNA-seq datasets). For each cell type, one gene with a local score ≥ 3 was selected for display to illustrate that each cell type has at least one strongly associated marker gene and that a gene which is strongly associated with one cell type is generally not strongly associated with most others. (**C**) Boxplot showing the distribution of local scores (GCAs) between matched (*n* = 174) and mismatched (*n* = 5678) gene-cell type pairs. The difference between these groups was assessed by calculating the Mann Whitney test p-value and effect size (r), along with the cohen’s D effect size. (**D**) Receiver operating characteristic (ROC) analysis demonstrating the ability of literature based GCAs to classify these manually curated matched versus mismatched gene-cell type pairs. The AUC was calculated as 0.903, and the sensitivity and specificity using a local score threshold of 3 are indicated in red.

**Figure 2 genes-12-00898-f002:**
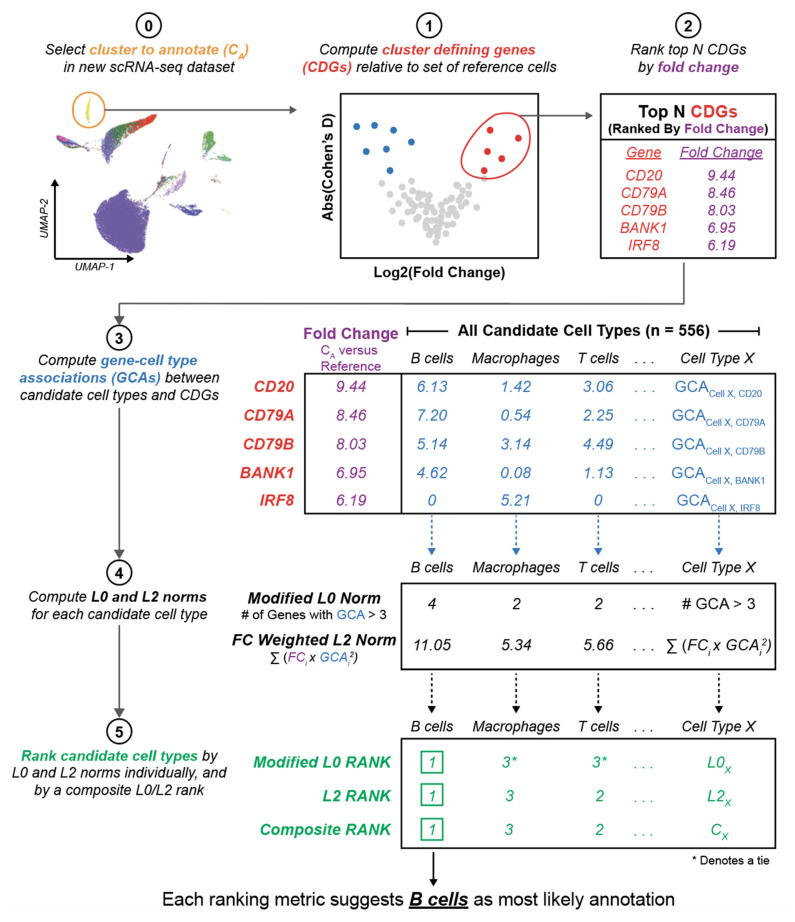
Schematic summary of the approach to leverage the literature knowledge graph to augment the annotation of cell type clusters in scRNA-seq datasets. This illustration presents the example of predicting the annotation of one cluster (B cells) from a previously published scRNA-seq dataset of the human kidney [40]. After clustering a new scRNA-seq dataset (step 0), we compute the top N cluster defining genes for a given cluster C_A_ relative to a defined set of reference cells (steps 1–2). The GCAs between each of these top N genes and all candidate cell types are computed (step 3), and the L0 and L2 norms are computed to summarize the level of literature evidence connecting these CDGs to each cell type (step 4). Candidate cell types are ranked by these norms (step 5), and the cell type showing the strongest association to the set of CDGs is selected as the most likely annotation for cluster C_A_. Note that only three candidate cell types are shown here for simplicity, but there were actually over 500 cell types considered (which mapped to 104 cell type priority nodes).

**Figure 3 genes-12-00898-f003:**
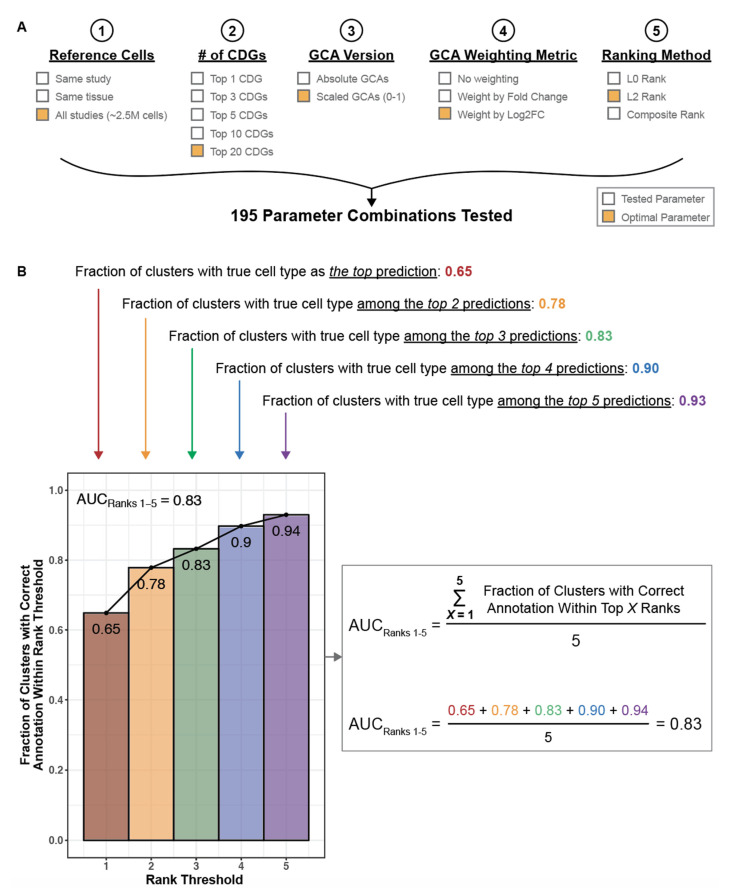
Hyperparameters tested for cluster annotation and schematic of algorithm performance evaluation. (**A**) Hyperparameters tested for each step of the cluster annotation algorithm, as described in Figure 2. The parameters that yielded optimal annotation performance are highlighted in orange. To compute CDGs, we compared the mean expression of all genes in the cluster of interest to their mean expression in a set of reference cells. Reference cells were taken as all other cells from the corresponding study (“within study”), all other cells from the corresponding tissue (“within tissue”), or all other cells from all processed studies (“pan-study”). After computing fold change values for each gene, we tested the selection of 1, 3, 5, 10, or 20 genes for the downstream annotation steps. We tested the use of absolute and scaled versions of GCAs (local scores between genes and cell types). To compute L2 norms, we tested the weighting of each GCA term with the corresponding fold change and log_2_FC values to increase the contribution of the strongest CDGs to the cell type prediction. To rank all candidate cell types, we considered a modified L0 norm (number of genes with GCA > 3 to the given cell type), an L2 norm, and a composite metric that considers both the modified L0 and L2 norms. (**B**) For each parameter combination (*n* = 195), a cumulative distribution plot was generated to illustrate the fraction of clusters which were correctly predicted within a given rank, ranging from rank 1 to rank 5. To summarize the performance, we considered the number of clusters which were annotated correctly (corresponding to the red bar at Rank Threshold = 1), and we estimated the area under this curve (denoted as AUC_Ranks 1–5_) as the average fraction of clusters for which the correct annotation was present among the top 1, 2, 3, 4, and 5 predictions.

**Figure 4 genes-12-00898-f004:**
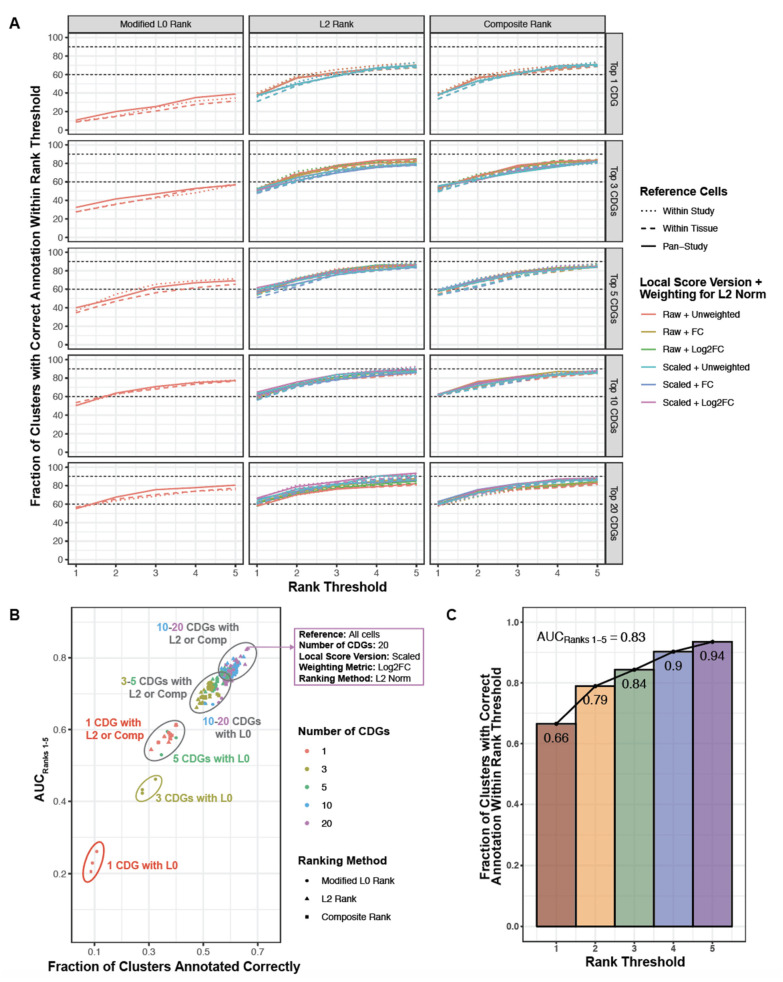
Performance evaluation of 195 parameter combinations for literature based augmented cell type annotation. (**A**) Each curve corresponds to one combination of parameters, where the parameters include: (1) cells used as reference to compute CDGs (encoded by line type and dot shape), (2) number of CDGs considered (encoded by the vertical facet variable), (3) the GCA version used (absolute or scaled; encoded by color), (4) the weighting method used in computing L2 norms of GCAs (also encoded by color); and (5) the metric by which candidate cell types were ranked to predict cluster identify (encoded by the horizontal facet variable). Each curve is generated from five points, specifically the percentage of clusters for which the correct annotation was present among the top 1, 2, 3, 4, or 5 predictions. (**B**) Performance summary of all 195 tested parameter combinations, considering the fraction of clusters which were annotated correctly (i.e., the top-ranked prediction which corresponded to the true cluster label) versus the AUC_Ranks 1–5_ metric. The metrics are highly, although not perfectly, correlated. The parameter combination which showed the highest fraction of correct annotations and the highest AUC_Ranks 1–5_ was taken as the optimal approach; these optimized parameters are shown in the purple inset. (**C**) Summary of algorithm performance with the optimized parameters as described in (**B**). The plot illustrates the fraction of clusters which were annotated correctly within the top 1, 2, 3, 4, or 5 ranks. The AUC_Ranks 1–5_ metric was estimated as the average of these five values.

**Figure 5 genes-12-00898-f005:**
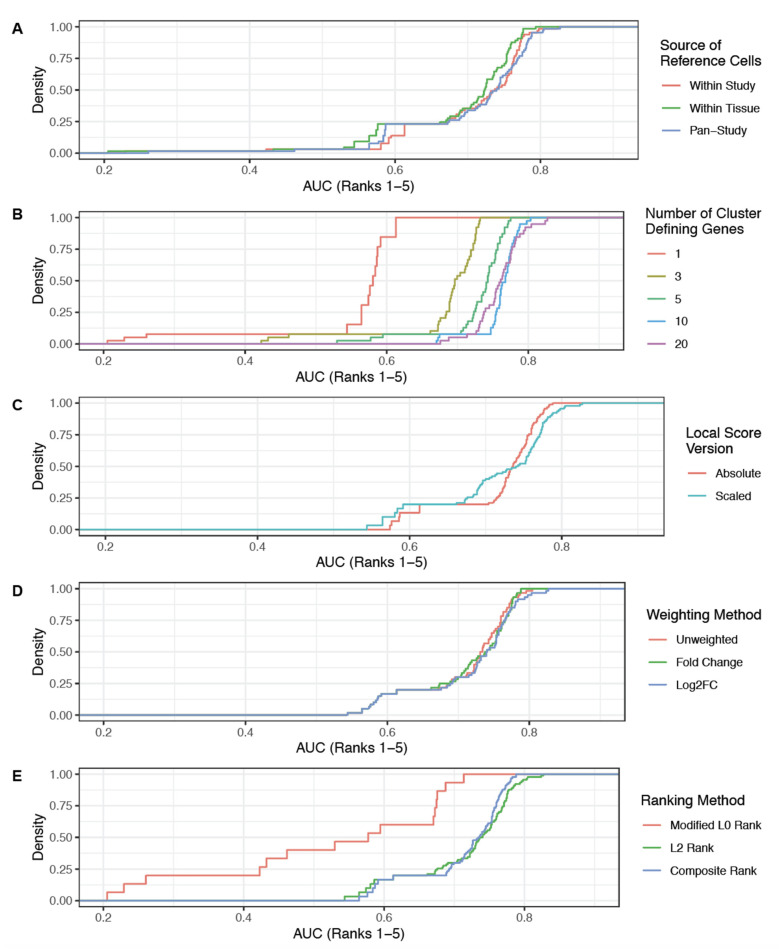
Effects of altering individual parameters on cluster annotation algorithm. Plots shown are the cumulative distribution functions of the AUC_Ranks 1–5_ metric. Each panel illustrates the effect of modifying a single parameter on the algorithm performance. (**A**) Modifying the source of reference cells used to compute CDGs for downstream analysis has a limited impact on performance, with the pan-study and within study options slightly outperforming the within tissue option. (**B**) Modifying the number of CDGs considered has a strong impact on performance, with fewer genes (e.g., 1, 3, or 5) showing considerably worse performance. (**C**) Modifying the local score version used (absolute or scaled) has a mixed effect, with scaled GCAs contributing more of the worst-performing (left tail) and best-performing (right tailed) parameter combinations. Note that only the parameter combinations which used an L2 norm-based rank or the composite rank were included in this analysis, as only absolute GCAs were considered for the modified L0 norm-based ranking. (**D**) Modifying the weighting method for calculating the L2 norm has minimal impact on performance, with weighting by either fold change (FC) or log_2_FC slightly outperforming the unweighted method. (**E**) Modifying the final ranking method has a strong impact on performance, with the modified L0 rank and L2-based rank showing the worst and best performances, respectively.

**Figure 6 genes-12-00898-f006:**
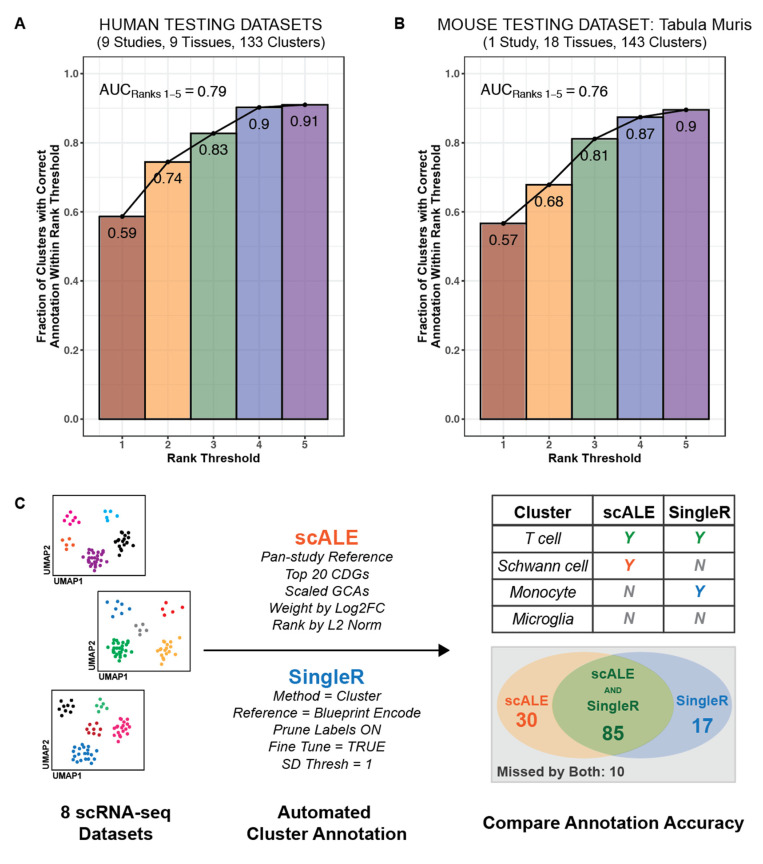
Application of scALE to testing datasets and comparison of performance with SingleR. (**A**) Using the previously optimized settings, scALE was applied to annotate 133 clusters from 9 “test” (i.e., previously unseen) studies [71,72,73,74,75,76,77,78,79]. (**B**) scALE was also applied to annotate 143 clusters from the murine scRNA-seq tissue atlas dataset, Tabula Muris [80]. (**C**) As a benchmark, we compared the accuracy of annotations from scALE to those obtained from SingleR [6] for 142 clusters from ten human studies [25,26,27,34,35,36,37,38,39,40]. scALE was implemented with the previously optimized parameters, and SingleR was implemented using the listed parameters. Manual review was performed to determine whether each cluster was labeled correctly by scALE only (*n* = 30), SingleR only (*n* = 17), both scALE and SingleR (*n* = 85), or neither (*n* = 10).

**Figure 7 genes-12-00898-f007:**
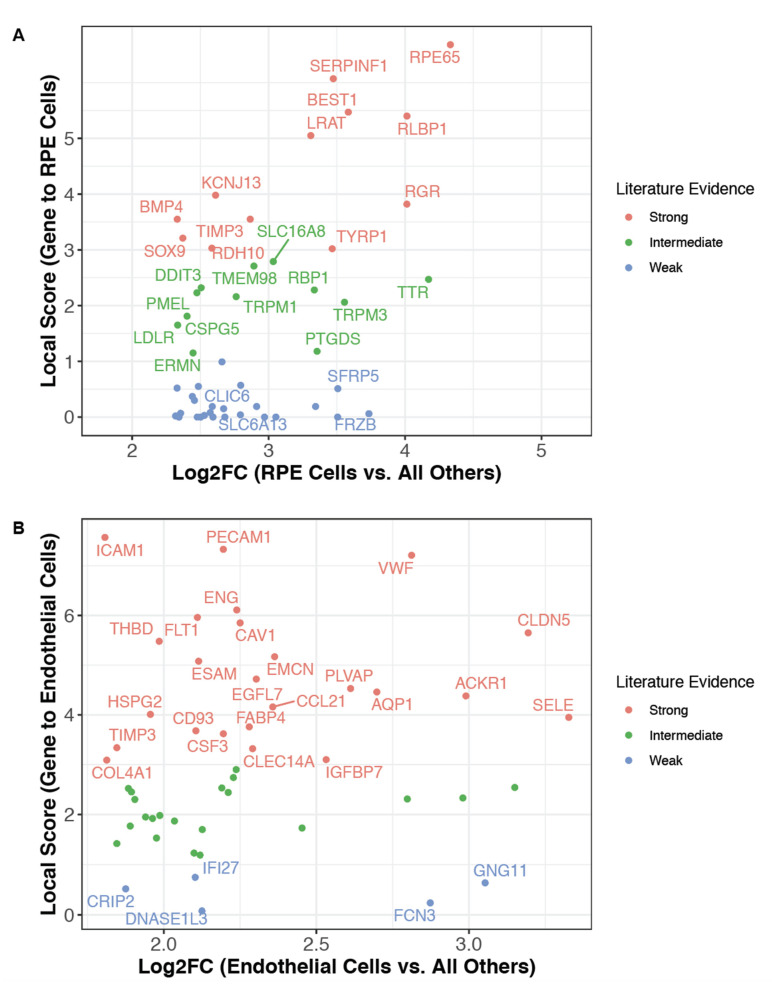
Literature-derived GCAs contextualize differentially expressed genes, highlighting poorly characterized markers of established cell types. (**A**) Transcriptional markers of retinal pigment epithelial (RPE) cells were identified by comparing the mean expression of all genes in RPE cells from two scRNA-seq studies [30,68] to their expression in all other cells from our reference datasets (see Appendix A). The top 50 markers (ranked by fold change) were classified for their level of literature association to retinal pigment epithelial cells as follows: strong: Local Score ≥ 3; intermediate: Local Score ≥ 1 and <3; weak: Local Score < 1. All genes with strong or intermediate evidence are highlighted by name, as are a subset of the genes with weak evidence that may warrant further evaluation. (**B**) The same process was applied as in (**A**), but for endothelial cells from 31 studies [26,27,28,30,33,35,36,37,38,39,40,42,43,46,47,48,49,50,51,52,53,55,56,62,64,67,72,73,74,75,76,78,79] rather than for RPE cells. Here, all genes with strong or weak evidence are highlighted by name.

**Table 1 genes-12-00898-t001:** List of studies that were used to tune the cluster annotation algorithm. The columns indicate (1) the tissue from which cells were derived in the study, (2) the title of the published study or dataset, and (3) the reference for the study [26,27,28,29,30,33,35,36,37,38,39,40,41]. The three pancreas datasets were integrated for one single analysis of cluster annotation.

Tissue	Study Title	Reference
Blood	Immune Cell Atlas: Blood Mononuclear Cells (2 donors, 2 sites)	Immune Cell Atlas
Blood	A single-cell atlas of the peripheral immune response in patients with severe COVID-19	PMID 32514174
Kidney	Spatiotemporal immune zonation of the human kidney	PMID 31604275
Liver	Resolving the fibrotic niche of human liver cirrhosis at single-cell level	PMID 31597160
Liver	A human liver cell atlas reveals heterogeneity and epithelial progenitors.	PMID 31292543
Lung	A cellular census of human lungs identifies novel cell states in health and in asthma	PMID 31209336
Lung	A single-cell atlas of the human healthy airways	PMID 32726565
Pancreas	De Novo Prediction of Stem Cell Identity using Single-Cell Transcriptome Data	PMID 27345837
Pancreas	Single-Cell Transcriptome Profiling of Human Pancreatic Islets in Health and Type 2 Diabetes	PMID 27667667
Pancreas	A Single-Cell Transcriptome Atlas of the Human Pancreas	PMID 27693023
Placenta	Single-cell reconstruction of the early maternal-fetal interface in humans	PMID 30429548
Retina	Single-cell transcriptomic atlas of the human retina identifies cell types associated with age-related macular degeneration	PMID 31653841
Retina	Single-cell transcriptomics of the human retinal pigment epithelium and choroid in health and macular degeneration.	PMID 31712411

**Table 2 genes-12-00898-t002:** List of human studies that were used to test the cluster annotation algorithm. The columns indicate (1) the tissue from which cells were derived in the study, (2) the title of the published study or dataset, and (3) the reference for the study [71,72,73,74,75,76,77,78,79].

Tissue	Study Title	Reference
Breast	Single-Cell Map of Diverse Immune Phenotypes in the Breast Tumor Microenvironment	PMID 29961579
Colon	Intra- and Inter-cellular Rewiring of the Human Colon during Ulcerative Colitis	PMID 31348891
Heart	Single-cell reconstruction of the adult human heart during heart failure and recovery reveals the cellular landscape underlying cardiac function	PMID 31915373
Joint (Knee)	Synovial cell cross-talk with cartilage plays a major role in the pathogenesis of osteoarthritis	PMID 32616761
Ovary	Single-cell reconstruction of follicular remodeling in the human adult ovary	PMID 31320652
Prostate	A Cellular Anatomy of the Normal Adult Human Prostate and Prostatic Urethra	PMID 30566875
Skin	Transcriptional Programming of Normal and Inflamed Human Epidermis at Single-Cell Resolution	PMID 30355494
Skin	Dissecting the multicellular ecosystem of metastatic melanoma by single-cell RNA-seq	PMID 27124452
Small Intestine (Ileum)	Single-Cell Analysis of Crohn’s Disease Lesions Identifies a Pathogenic Cellular Module Associated with Resistance to Anti-TNF Therapy	PMID 31474370

## Data Availability

The datasets used to test the methods presented were accessed from the NCBI Gene Expression Omnibus (GEO) or another public data repository (see Table 1, Table 2, and Appendix A). These datasets, in the processed and annotated form as used in our analyses, can be accessed and downloaded by academic researchers from academia.nferx.com and will be made accessible to non-academic researchers upon reasonable request. All newly generated data used in the cluster annotation workflow (e.g., cell graph, cell type mapping to priority nodes, raw and scaled GCA matrices) are available in Appendix A. The described annotation workflow has been wrapped into the python package scALE. Package documentation and associated files can be accessed in this publicly accessible Google Drive folder: https://drive.google.com/drive/folders/1_ApNN6hoekmhVCSbbcQg095_O3Mhje-f?usp=sharing. This includes a README file and a Jupyter notebook that illustrates the use of scALE, including an example of how the annotation predictions were generated for this manuscript. To install scALE, prospective users first need to register for an account at academia.nferx.com in order to obtain a user name and password; the API key, which is required for package authentication, can then be found on the website as illustrated in the tutorial at this link: https://drive.google.com/file/d/1w4Kk9nWyME48rr-3C23pVVH4HPnQHT1Y/view?usp=sharing. The source code is downloaded to the python environment’s site-packages directory when the package is installed via pip. scALE can also be implemented through a user interface in the Single Cell section of academia.nferx.com (https://academia.nferx.com/dv/202007/singlecell/?view=scale) by uploading data in one of the following formats: (i) a gene expression matrix along with a metadata file containing cluster assignments, (ii) a table of average gene expression values in each cluster, or (iii) a table of pre-computed CDGs. The required formats for file uploads are explained on the website, and example files for each mode (matrix with metadata, table of average expression values, or table of CDGs) can be downloaded from the website or from this Google Drive folder: https://drive.google.com/drive/folders/12V5YY4rruBBFp7tixJxfqcYOdDTAd3mI?usp=sharing.

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
