# Peer review of "A Literature-Derived Knowledge Graph Augments the Interpretation of Single Cell RNA-seq Datasets"

_genes, 2021, doi:10.3390/genes12060898_

Round 1
Reviewer 1 Report
Overview
The authors present a novel framework for leveraging the biomedical literature in order to score genes regarding their association with known cell types and to then use these genes to annotate cell types in scRNA-seq data. The authors perform a comprehensive evaluation into the many parameters and variants of their method. With the best-performing variant, the authors demonstrate that their method is able to produce annotations with reasonable accuracy on a comprehensive collection of datasets from diverse tissues. Moreover, the authors demonstrate that by using the literature, they are able assess the literature support for genes that are highly expressed in specific cell type populations.
In general, this paper is very comprehensive and its methods are clearly described. I found the schematic diagrams illustrating their methods, experiments, and evaluation metrics to be very helpful for making their detailed study very comprehensible. The authors demonstrate that leveraging the literature is a promising direction for improving upon the important tasks of characterizing cell type-specific marker genes and for performing cell type annotation. My biggest concern about this work regards their method for parameter tuning and evaluation. Specifically, it seems that the authors are tuning the parameters on the same data that they use for evaluating their method, which will tend to produce overly optimistic results. My second concern regards comparison to existing approaches for automated cell type annotation. While the author’s goal is not necessarily to compete with existing approaches, rather they seek to complement them, it would be beneficial to compare how their literature-based approach compares to current state-of-the-art methods for automated cell type annotation. If their method performs much worse, then it would perhaps not be a valuable approach for verifying the results of an existing automated pipeline. Please see my detailed concerns outlined below.
Major comments:
- It is not clear which datasets the authors used for performing the parameter tuning. Did the authors test their method on the same data for which they used to optimize their parameters? If so, this would lead to overly optimistic results and the authors should make sure to tune their parameters on separate data from which they use to evaluate their method.
- The authors demonstrate that their literature-based method is able to produce reasonably accurate automated cell type annotations; however, it is not clear how these results compare to existing state-of-the-art approaches. Granted, the authors make clear that their goal is not necessarily to compete with the state-of-the art, rather, they note that their tool “should be viewed as a tool for augmenting current annotation workflows rather than as a standalone automated pipeline to replace other methods.” Nonetheless, it would help put these results into context with a comparison to state of the art methods for automated cell type annotation such as those provided by SingleR (https://doi.org/10.1093/bioinformatics/btz292), scMatch (https://doi.org/10.1093/bioinformatics/btz292), or others. Does the author’s literature-based approach perform worse than current automated approaches? If so, it may not work to verify the output of automated approaches.
- In Section “The literature knowledge graph highlights uncharacterized markers of established cell types”, the authors state, “By contextualizing each CDG for the novelty, or lack thereof, of its association with the given cell type, our literature knowledge graph also enables researchers to rapidly identify novel markers of even well studied cell types.” The results in this section do not seem to support this statement. The literature-derived knowledge graph does not enable researchers to find new marker genes per se, rather, the authors show that it enables researchers to assess the literature support of genes that ~appear~ to be marker genes in their target dataset. While I imagine that this will be a useful feature, it is an overstatement to say that their knowledge graph enables the discovery of new marker genes.
- On Line 77, the authors state, “We merged this manually curated cell graph with the 77 EBI Cell Ontology graph [19–21] by mapping identical nodes to each other and preserving all parent child relationships documented in each graph.” Why did the authors not simply start with the Cell Ontology? It would be helpful to have an explanation on why an additional curated ontology graph was necessary.
- The authors state, “all code used to perform and evaluate 745 cluster annotations will be made available on github.” This code should be made available prior to publication. Also, will there be an open-source software tool for users to run the methods described in this manuscript?
Minor comments:
- While the comprehensive description and evaluation of all of the parameters and variants of their methodology is valuable, especially for those who may wish to utilize this method, it would improve readability of the manuscript to include in the main text’s Methods section only a description of the best-performing algorithm and parameters (that is, the parameters outlined in Figure 3A) and move the description of the other variants and their performance results to the supplement. While not strictly necessary, this would greatly help focus the paper’s message on demonstrating the author’s thesis that, “the systematic application of a literature derived knowledge graph can expedite and enhance the annotation and interpretation of scRNA-seq data.”
- In Line 138, the authors state, “we first curated a set of cell type defining genes, i.e. genes which were used to identify 139 cell types in previously published manually annotated scRNA-seq datasets [25–32].” Did the authors consider using the CellMarker database (https://doi.org/10.1093/nar/gky900), which is a manually curated database of cell type marker genes? If so, was there a reason they did not use it? It not, the authors should cite this work and discuss its relevance to the authors’ manual curation process.
- Figure 1 is first referenced in the text after Figures 2 and 3, which made it a bit confusing to follow. Ideally, Figure 1 should be referenced first, in the methods section.
- In the caption for Figure 1, the authors provide a matrix of genes and cell types. Why were these specific genes selected to display? Is this a random selection? Furthermore, the authors state, “These genes and cell types were extracted from a 426 set of scRNA-seq datasets.” If there are not too many references, it would help if these references are included here in the caption.
Reviewer 2 Report
In this article Doddahonnaiah et. al. utilize the NLP model previously developed by their group to create a gene-to-cell association (GCA) score for human protein-coding genes and cell types, and demonstrate the utility of these scores in automating the annotation of cell types in scRNA-seq data. The approach is novel and potentially highly useful as it could simplify this tedious step in scRNA-seq workflows. I have a few comments on the description of their methods and results.
1) Authors should clarify how the composition of the biomedical corpus used to create the GCA scores affects the annotation. Specifically, 1) In some cases a GCA could be species specific. Do the authors check for species composition in their corpus or take this into account? 2) Are GCA scores stronger for certain cell types vs the others due to them being over-studied (more literature available)? For example, looking at the Fig 1B it seems that immune cells and their corresponding genes get a stronger GCA-score in their tested scRNA-seq dataset, this could imply that the accuracy of annotation using this model is higher for immune cell types. One can check this by plotting the GCA-score distribution for top 100 Genes per cell type.
2) It's not mentioned in the methods section how the filtering of cells and genes was done for the scRNA-seq datasets before normalization and clustering. Poor filtering could affect the clustering and annotation.
3) Authors mention 103 “priority nodes” capture major cell types, but from the supplementary file 2, there seems to be 144 unique entries for "mapped priority nodes".
4) To demonstrate the potential uncharacterized markers for Figure 7, the authors identified the top 50 markers by looking at gene fold changes compared to the "pan study" reference. As noted by authors elsewhere, the batch effects could influence this result. Do the authors also identify potentially novel/understudied markers if they use the "same study" reference to calculate the top marker genes? Basically I wonder how specific these potentially novel markers are to the RPE/Endothelial cells.
5) Finally, It should be indicated how the authors plan to make their annotation algorithm/platform available for use.
Round 2
Reviewer 1 Report
I appreciate the thoughtful and thorough response from the authors that addressed many of my significant concerns. Most significantly, tuning the algorithm on separate data from which it was tested greatly solidified the scientific soundness of this study. Furthermore, the author’s comparison to SingleR on 9 independent studies further supports its utility in the cell type classification task.
Unfortunately, I still have a significant concern regarding the reproducibility of this study without publicly available source code. In order for this study to be acceptable for publication in a scholarly journal, it is necessary that the code be made available to the scientific community. The authors sent a separate correspondence to the reviewers via the editorial office explaining that the source code implementing the methods and analysis presented in this study would not be made publicly available on GitHub. Without source code accompanying the manuscript, the reproducibility of this study is greatly diminished (https://www.ncbi.nlm.nih.gov/pmc/articles/PMC8144864/).
Lastly, it appears that there is a “floating figure” in between Figures 5 and 6 that does not have an associated caption. This appears to be Figure 6 from the first version of the manuscript that displays UMAP plots for cell types from retina, blood, and pancreas datasets. Is this floating figure supposed to be there in the manuscript? If so, I suggest changing this figure to show UMAP plots demonstrating how their method performed on the separate 9 test sets rather on datasets that were used to tune the algorithm.
